# Docosahexaenoic Acid Delivery Systems, Bioavailability, Functionality, and Applications: A Review

**DOI:** 10.3390/foods11172685

**Published:** 2022-09-02

**Authors:** Wenwen Lv, Duoxia Xu

**Affiliations:** 1Beijing Advanced Innovation Center for Food Nutrition and Human Health, Beijing Technology and Business University, Beijing 100048, China; 2School of Food and Health, Beijing Technology and Business University, Beijing 100048, China; 3Beijing Engineering and Technology Research Center of Food Additives, Beijing Technology and Business University, Beijing 100048, China; 4Beijing Higher Institution Engineering Research Center of Food Additives and Ingredients, Beijing Technology and Business University, Beijing 100048, China; 5Beijing Key Laboratory of Flavor Chemistry, Beijing Technology and Business University, Beijing 100048, China; 6Beijing Laboratory for Food Quality and Safety, Beijing Technology and Business University, Beijing 100048, China

**Keywords:** DHA delivery systems, digestion, bioavailability, anti-obesity, food applications

## Abstract

Docosahexaenoic acid (DHA), mainly found in microalgae and fish oil, is crucial for the growth and development of visual, neurological, and brain. In addition, DHA has been found to improve metabolic disorders associated with obesity and has anti-inflammatory, anti-obesity, and anti-adipogenesis effects. However, DHA applications in food are often limited due to its low water solubility, instability, and poor bioavailability. Therefore, delivery systems have been developed to enhance the remainder of DHA activity and increase DHA homeostasis and bioavailability. This review focused on the different DHA delivery systems and the in vitro and in vivo digestive characteristics. The research progress on cardiovascular diseases, diabetes, visual, neurological/brain, anti-obesity, anti-inflammatory, food applications, future trends, and the development potential of DHA delivery systems were also reviewed. DHA delivery systems could overcome the instability of DHA in gastrointestinal digestion, improve the bioavailability of DHA, and better play the role of its functionality.

## 1. Introduction

Docosahexaenoic acid (DHA), mainly extracted from microalgae and fish oil, is one of the most important omega-3 polyunsaturated fatty acids (PUFAs) and has significant health benefits [1,2]. DHA is known to be essential for the growth and development of the infant brain. DHA has been found to positively affect diseases such as atherosclerosis, myocardial infarction, diabetes mellitus, arthritis, heart disease, and cancers. Among them, chronic non-communicable diseases such as cardiovascular diseases, diabetes, and some cancers are associated with obesity [3]. In the last 30 years, the prevalence of obesity has almost doubled, and obesity will lead to an increased risk of these diseases, a worldwide public health problem [4]. Indeed, DHA has been found to improve metabolic disorders associated with obesity [5,6,7]. DHA has different anti-inflammatory, anti-obesity, and anti-adipogenic effects in some identifiable human, animal, or cell models [8,9,10].

However, DHA cannot be synthesized directly in the human body and needs to be obtained from food. DHA can also be converted by enzymatic reactions of α -linolenic acid [11]. As we know, DHA has the scent of fish flavor. In addition, DHA is difficult to dissolve in water, sensitive to light, heat, oxygen, and metal ions, and easy to degrade, resulting in flavor changes and inactivation due to the polyunsaturated double bond (Figure 1). Therefore, it is an urgent problem to develop delivery systems to mask the smell of fish, maximize the retention of DHA activity, and improve its stability and utilization in food [7,12].

Accordingly, this review mainly focused on the various delivery systems of DHA and in vivo and in vitro digestion, as well as the functionality of DHA delivery systems. In addition, the application of DHA delivery systems in food was also discussed. The review reported the future research direction and development potential of DHA delivery systems in food. 

## 2. Delivery Systems of DHA

A delivery system is an efficient method to increase the intake of DHA and improve its bioavailability. Different delivery systems have different structures (Figure 2), both committed to protecting DHA from adverse storage environments and improving its stability and bioactivity. This section provides a detailed overview of the different delivery systems for DHA (Table 1). The preparation, characterization, and challenges of different DHA delivery systems were introduced.

**Table 1 foods-11-02685-t001:** Summary of DHA delivery systems.

DHA Delivery System	Materials	Size	Encapsulation Efficiency	Storage Stability	Main Factors Affecting Stability	Reference
Microemulsions	Tween 80, CaCl_2_, surfactin	15–50 nm	N/A	Stable for 2 years at 4 °C	Surfactant	[13]
Nanoemulsions	Tween-40	10–30 nm	N/A	Stable over 100 days at 4 °C	Preparation technology	[14]
	Coffee oil, algae oil, Span 80, Tween 80, water	30 nm	100%	Emulsions were stable when heated up to 110 °C at a pH 6	N/A	[15]
Multilayered emulsions	Lecithin, chitosan, maltodextrin	N/A	N/A	Stable for 12 days at 30 °C and 60 °C	The composition of the emulsions	[16]
Liposomes	L-α-Phosphatidylcholine	129.6 ± 0.4 nm	70.3 ± 1.0%	N/A	N/A	[17]
Pickering emulsions	Water, gelatin	2.11–34.68 μm	N/A	Stable for 3 days at 4 °C, room temperature, 37 °C	Solution pH, homogenizing time, homogenizing speed, storage temperature	[18]
Nanoparticles	PLGA, chitosan	145–341 nm	80.45%	Stable for 42 days at 30–80 °C	Materials	[19]
	Zein and PLGA	319.9 ± 8.28 nm	84.6%	Stable over 35 days at 4 °C	Materials	[20]
Microcapsules	Casein, glucose, lactose	14.173 μm	98.66%	Stable for 8 weeks at 45 °C	Wall materials	[21]
	Dodecenyl succinic anhydride-esterified agarose	100–400 μm	65–85%	Stable for 30 days at room temperature	N/A	[11]
Gels	Water, gelatin	1.81 ± 0.02 mm	N/A	Stable length of study	Forms	[22]

**Figure 2 foods-11-02685-f002:**
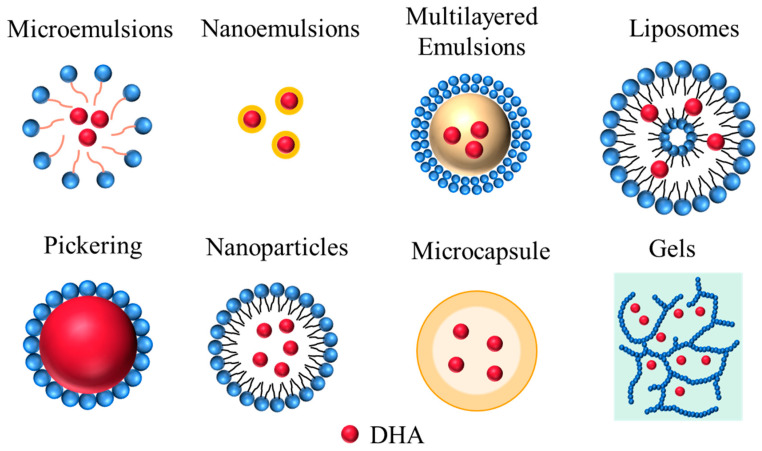
Depiction of different delivery systems indicating the possible location of DHA.

**Table 2 foods-11-02685-t002:** Functionality of DHA in delivery systems.

Functionality	DHA Delivery System	Results	Reference
Improve cardiovascular diseases	Microemulsions	Increased DHA bioavailability by 77% and 41% in the heart and brain lipids	[23]
Improve visual and neurological/brain development	Nanoparticles	Enhanced the DHA content in the brain	[24]
	Microcapsules	Increased DHA levels in blood	[25]
Improve diabetes mellitus	Microemulsions	Increased the absorption of DHA, which could reduce oxidative stress induced by high blood glucose	[26]
Anti-obesity	DHA-PC	Reduced liver weight and hepatic triglyceride levels in OLETF rats to reduce obesity-induced fatty liver	[27]
Anti-inflammation	DHA	Changed the secretion of adipokine in 3T3-L1 cells and had an anti-inflammatory effect	[28]
	DHA	Decreased TNF-α, IL-1β in LPS-induced inflammation and mediated anti-inflammatory effects through the NF-κB signaling pathway	[29]
	DHA	Enhanced anti-inflammatory IL-10 secretion and significantly inhibited the expression of IL-6, IL-1β, TNF-α in macrophages	[30]

### 2.1. DHA Microemulsions

Microemulsions are thermodynamically stable transparent systems consisting of surfactant, oil, and water phases [31]. The size of microemulsions was typically less than 100 nm, usually distributed between 10 and 50 nm [32,33]. The properties of microemulsions provide opportunities and potential benefits for incorporating DHA into food and beverage products. Furthermore, the bioavailability of DHA in microemulsion was improved. Microemulsions have been proposed as efficient carriers for various active molecules [17].

Amphiphilic molecules are capable of reducing the interfacial tension between oil and water and increasing the steric hindrance and electrostatic repulsion between micelles, which can stabilize DHA microemulsion [34]. Surfactants, such as sodium dodecyl sulfate, lecithin, whey protein, Tweens, chitosan, and lipoid, are often used as amphiphilic molecules to prepare DHA microemulsions. Sugasini and Lokesh [23] produced microemulsions with chitosan, acacia gum, whey protein, and lipoid, respectively, which could induce a significant increase in the stabilization and bioavailability of DHA in fish oil microemulsions. Because the lipophilic molecules in the microemulsions can not only effectively cross the intestinal barrier but can also cross the blood-brain barrier, which would be more beneficial for DHA absorption and utilization. Due to the maturity of microemulsion technology, the materials and methods to prepare microemulsion have experienced gradual innovation. Surfactin, as a natural peptide biosurfactant, has powerful surface activity. DHA single cell oil (DHASCO) microemulsions prepared with surfactin could greatly improve its physical and antioxidant stabilities. The size of DHASCO microemulsion with 0.2 mM surfactin was decreased from 140 nm to 15 nm [13]. Xu et al. [17] constructed co-surfactant-free DHA microemulsions prepared using ultra-high temperature and high-temperature short time. The combination of these two techniques could show a further increase in DHA thermal stability.

### 2.2. DHA Nanoemulsions

Similar to microemulsions, nanoemulsions are also composed of water, oil phases, and surfactants (emulsifiers/stabilizers) in appropriate proportions. Unlike microemulsions, the preparation of nanoemulsions also needs structural modifiers, weighting agents, and ripening agents [35,36,37]. Additionally, the particle size of nanoemulsions is generally between 10–100 nm [15,38]. Compared with traditional emulsions, nanoemulsions are advantageous in terms of their small particle size, improved solubility, and excellent physical and storage stability [39]. Currently, nanoemulsions can be fabricated using high-energy and low-energy emulsifications. Low-energy emulsification is the spontaneously formed small oil droplets under specific conditions, while high-energy emulsification utilizes strong mechanical forces capable of breaking up the oil and water phases, leading to tiny oil droplets. High energy emulsification is widely used in foods due to its precise particle size control and easy scale-up processes [40]. Nanoemulsions are extensively used to deliver fat-soluble ingredients, including fish oil, algae oil, and canola oil. Nanoemulsions have higher stability and bioavailability in food than microemulsions [41].

The physical stability of nanoemulsions under different environmental conditions is very important for DHA application in food processing. Physical instabilities include phase separation, sedimentation, flocculation, and coalescence [42]. Among them, an emulsifier is an essential part of nanoemulsion formation, making nanoemulsion formation easier to achieve. Tween-40 (T-40), sodium caseinate (Na-CA), and soybean lecithin (SL) are commonly used as emulsifiers in nanoemulsions. DHA nanoemulsions were prepared by microfluidizer with different emulsifiers such as T-40, Na-CA, and SL. The average particle size of T-40 nanoemulsion (148 nm) was smaller than that of NA-Ca (206 nm) and SL (760 nm). Structural features of T-40 nanoemulsion remained stable during storage, whereas nanoemulsion prepared with Na-CA caused flocculation and coalescence. It was also found that DHA nanoemulsion prepared with T-40 had higher storage stability and bioavailability [43]. Hence, Karthik and Anandharamakrishnan [14] fabricated a DHA nanoemulsion with T-40, which remained intact during the emulsification process, and no structural change was observed in DHA nanoemulsion. DHA nanoemulsions prepared by high-speed and high-pressure homogenization displayed good physical and chemical stabilities, and the bioavailability of DHA was also enhanced. Self-nanoemulsifying drug delivery systems (SNEDDS) are also an effective way to improve the bioavailability of DHA. SNEDDS have been reported to increase the oral absorption of lipophilic drugs [44]. Singh et al. [45] prepared DHA-SNEDDS by employing different combinations of the oil phase, surfactants, and cosurfactants. The optimized formulation of DHA with SNEDDS-based nanoemulsion has a particle size of 17.6 ± 3.5 nm, which enhances the dispersibility and bioavailability of DHA.

### 2.3. DHA Multilayered Emulsions

Monolayer emulsion was produced by adsorbing oil droplets with emulsifiers during the homogenization process and producing a protective layer to prevent oil droplets from accumulating. The multilayered emulsion is always made up of an emulsifier, polyelectrolyte, and the layer-by-layer technique. During the preparation of multilayered emulsion, an ionic emulsifier is added to stabilize the oil in water phase. Then, a polyelectrolyte of opposite charge is added to encapsulate the emulsion droplets, resulting in a two-layer system of emulsions coated with oil particles. The multilayered emulsions could be achieved by repeating the process [46,47]. The stability of multilayered emulsions was higher than that of monolayer emulsions. It has been proven that multilayered emulsions prepared with chitosan and lecithin increased the oxidative stability of DHA in comparison with monolayer emulsions [16,48]. The multilayered emulsions have better stability than monolayer emulsions in the process of emulsion preparation and storage, because multilayered emulsions have smaller particle sizes and could form a thicker interfacial layer than monolayer emulsions [49].

### 2.4. DHA Pickering Emulsions

Pickering emulsions are stabilized by solid particles using colloidal particles ranging in size from a few nanometers to hundreds of microns [50,51]. Solid particles could be irreversibly adsorbed at the oil-water interface to form stable Pickering emulsions. Pickering emulsions are generally considered highly stable because the interfacial adsorption of the particle stabilizers is almost irreversible [52,53]. The solid particles could be irreversibly adsorbed to the oil-water interface as a physical barrier, which could provide high stability for DHA and reduce the oxidation of DHA. It could also protect DHA from the adverse effects of external environmental conditions [54]. Further, due to the high energy desorption of particles from the interface, Pickering emulsions have good stability and could effectively protect DHA for a long time [55,56]. Pickering emulsions could be easily added to foods directly and evenly dispersed into foods, especially beverages [18]. Therefore, DHA Pickering emulsions have received more and more attention in recent years.

Colloidal particles are key components in the formation of Pickering emulsions. Food-grade colloidal particles could be prepared from polysaccharides, protein, fat crystals, and other substances such as wax and flavonoids [51]. However, there was a limited study on the food-grade DHA Pickering emulsions [57,58]. Zhou et al. [59] constructed DHA-rich algal oil Pickering emulsions with gliadin/chitosan colloidal particles. The colloidal particles were adsorbed and anchored on the oil-water interface and used as building blocks to protect DHA, endowing DHA Pickering emulsions with viscoelastic and self-supporting properties. Thus, the liquid algal oil was converted into a soft solid to prevent its oxidation and improve its stability. Ding et al. [18] prepared DHA Pickering emulsions with gelatin nanoparticles via the homogeneous method. The particle sizes decreased linearly with the increase of pH and homogenization time, and exponentially decreased with the increase of homogenizing speed. Therefore, DHA Pickering emulsions could improve DHA stability and prevent its oxidation, which has a potential application prospect in the food industry.

### 2.5. DHA Liposomes

Liposomes are tiny spherical particles with bilayer structures and particle sizes ranging from nanometer to micrometer, which are small membrane bubbles composed of lipids with hydrophobic and hydrophilic phases. Non-polar domains with hydrophobic biological activity in liposomes are located between bilayers formed by surfactants, while polar domains are located in the water interior of liposomes. Surfactant bilayer could be made from natural ingredients such as phospholipids, phosphatidylcholine, and synthetic ingredients like Tweens [60]. Liposomes play an important role in the encapsulation of bioactive compounds [61], which are mainly used to improve the sustainable release and stability of bioactive compounds [62,63].

The oxidative stability of DHA could be improved, and the strong bad odor of DHA could be masked significantly in a liposome, which showed great potential in the application of functional products [64,65,66]. Shirouchi et al. [67] reported that coating DHA in phosphatidylcholine (PC) liposomes significantly improved the oxidative stability of DHA. Kazuhiro, Seiji, and Morio [68] proved that coating DHA with phosphatidylethanolamine (PE) in liposomes increased its stability and resistance to lipid peroxidation. Further, nanoliposome technology is more advanced and efficient due to its protection of sensitive bioactive molecules, storage stability, high loading capacity, enhanced bioavailability, and sustained-release mechanism, which could improve the solubility and bioavailability of DHA [64,69]. DHA liposomes could promote the application and development of DHA in many fields, such as food and medicine.

### 2.6. DHA Nanoparticles

Nanoparticles are a material with a particle size from 10 to 1000 nm [70]. Compared with other delivery systems, nanoparticles have many advantages [71,72], which are widely used in food, medicine, and other fields [73,74,75]. The nanoparticle technology in the functional food field mainly focuses on the delivery of active substances such as lutein, astaxanthin ester, and curcumin [76]. In recent years, there have been some studies on the DHA nanoparticle. Nanoparticles could be formed by adding DHA, pre-dissolved in ethanol, to casein solutions. The nanoparticles had a significant protective effect on DHA and showed good colloid stability and bioactivity [77]. Liu et al. [19] prepared the DHA-loaded nanoparticle with polylactic acid (PLA) and chitosan. The nanoparticles had a better encapsulation rate (80.45%) and higher-water solubility than free DHA. Further, the encapsulation rate of DHA-loaded nanoparticles via zein/poly (lactic-co-glycolic) acid stabilized nanoparticles was up to 84.6%, which also showed good stability after 35 days of storage. The possible reason was that the hydrogen bonds and electrostatic interactions between PLA and chitosan/zein made the structure more compact and avoided DHA oxidation. The water solubility of DHA nanoparticles was 750 times that of free DHA, which may be because the hydrophilic heads (-COOH and -OH) of PLA significantly increased the water solubility [20].

Low-density lipoprotein (LDL) is a natural macromolecular assembly responsible for delivering cholesterol and other lipids in plasma [78]. Due to the bioactivity of DHA, LDL was recombined with DHA to form nanoparticles that were expected to treat cancer. Recombinant LDL with natural lipids could produce stable LDL particles with high DHA payloads [24]. Interestingly, LDL reconstructed from unesterified DHA exhibited higher physicochemical stability and selective anti-cancer cytotoxic activity. Mulik et al. [79] analyzed the structure of the nanoparticles. They found that unesterified DHA preferentially added to the outer surface of LDL, where the anionic carboxyl terminal of DHA was exposed to the LDL surface, gave the surface of the nanoparticles a negative electric charge. The negative surface charge promoted monodispersion and uniform distribution of LDL-DHA nanoparticles. Further structural analysis by cryogenic electron microscopy showed that LDL-DHA nanostructures were surrounded by a phospholipid bilayer with the aqueous core, which showed improved physical and chemical stabilities. Other studies have found that LDL-DHA nanoparticles are cytotoxic to both rat hepatocellular carcinoma and human hepatocellular carcinoma, which showed good therapeutic potential [80,81]. DHA nanoparticles have a good application prospect in the field of disease treatment.

### 2.7. DHA Microcapsule 

Microencapsulation technology could protect the oxidation of unsaturated fatty acids, extend the shelf life, and make them controlled release at a specific time and appropriate location. Microencapsulation is one of the most commonly used techniques to improve the oxidation stability and bioactivity of DHA [82]. Spray drying, spray cooling, complex coacervation, and polymerization are ordinarily used to explore the microencapsulation of different food ingredients [83]. Although spray drying is the most commonly used method for DHA microencapsulation, the powder particles developed by spray drying are easily oxidized [84,85,86]. At present, the spray cooling process and complex coacervation are more often used to prepare DHA microcapsules. Xiao et al. [11] prepared DHA microcapsules with dodecenyl succinylated agarose as raw material by spray cooling method, which exhibited excellent oxidation stability and good release characteristics under simulated gastric and intestinal conditions. Lu et al. [87] used a complex coacervation method to encapsulate DHA microalga oil in whey protein isolate arabic gum, which improved the physical and oxidative stabilities of DHA. With the development of microcapsule technology, new microcapsule preparation technology has been developed. Yildiz et al. [88] prepared DHA microcapsules using the pH-displacement-thermoultrasonic method, which could further improve the stability of DHA and produce capsules with higher oxidative stability.

The protection and controlled release of microcapsules mainly depend on the encapsulation wall materials, which have a significant impact on encapsulation efficiency, oxidative stability, size, shape, density, and water content [89,90,91]. Therefore, the choice of wall material is essential for preparing microcapsules with excellent performance. Carbohydrate (maltodextrin, modified starch, etc.), protein (gelatin, casein, milk protein, and soy protein), and biopolymers (mainly Maillard reaction products) are the three main categories of wall materials for the preparation of microcapsules [92,93]. Studies have reported that single wall material could not present all the desired characteristics, and using single wall material may have some disadvantages, such as low emulsification capacity and high cost of treatment. Therefore, it is optimum to use a mixture of different proportions of carbohydrates, proteins, gums, and other substances in varying proportions as wall material. Embedding DHA with polymer blends could achieve higher encapsulation efficiency and lower cost than using single wall material [94,95,96,97]. Vaziri et al. [98] made DHA microcapsules by mixing alginate, pectin, and gelatin, which improved the bioavailability and storage stability of DHA. Augustin, Sanguansri, and Bode [99] found that the Maillard reaction products obtained by heating protein and carbohydrates had a positive effect on the encapsulation efficiency of DHA-rich fish oil. At present, the physical structure of microcapsules and the chemical anti-oxidation effects of Maillard reaction products on the stabilization of DHA in the microcapsules remain unclear, which could be studied in the future.

### 2.8. DHA Gels

Gel is a typical semi-solid food system, a spatial network structure formed by polymer solution or sol connected to each other under certain conditions. In recent years, proteins such as gelatin, soy protein, and polysaccharides such as starch, carrageenan, pectin, and flaxseed gum have often been used as gel substrates [100,101,102]. Gels have the forms of microgel, hydrogel, emulsion gel, and oleogel. Due to the diversity of structure and composition, gels have wide application prospects in the food industry, such as fat reduction, probiotic release, and flavor control [103,104,105]. However, few studies have been done on DHA gels. Haug et al. [22] found that embedding fish oils in gel could improve the bioavailability of DHA. Tolasa, Chong, and Cakli [106] added DHA-rich omega-3 fatty acids to the surimi gel system and observed that the high uniform dispersion and chemical stability of omega-3 fatty acids could be achieved in the surimi gel system without the use of antioxidants. It suggested that surimi gel could be used as a protein-based carrier to develop seafood products with high omega-3 fatty acids. The stability and bioavailability of DHA could be improved with a gel delivery system, which could also be very promising for DHA applications.

## 3. Digestion of DHA in Delivery Systems

Digestion involves many different processes that convert food into absorbable nutrients in the body [107]. Digestion begins orally, where food is chewed and mixed with saliva. Saliva contains amylase, which hydrolyzes starch into small particles [108,109]. Food pulls through the oral and esophagus into the stomach. The physical processing of food in the stomach is relatively inactive compared to oral. The stomach is essentially a storage vessel that controls the release of nutrients into the small intestine, which is the main site for the digestion and absorption of nutrients. In vitro and in vivo digestion models were usually studied in the digestion of DHA in delivery systems.

### 3.1. In Vitro Digestion

DHA delivery systems could improve the stability of DHA. At the same time, the bioaccessibility and bioavailability of DHA in delivery systems are more important. In vitro simulated digestion has been widely used to study the gastrointestinal behavior of food, from which the bioaccessibility of DHA could be determined (Figure 3). The in vitro simulated digestion has the advantage of high speed, low cost, less labor intensiveness, and no ethical restrictions [110]. In vitro simulated digestion mainly includes static digestion models, semi-dynamic digestion, and dynamic digestion models (Figure 4). In vitro digestion of DHA in delivery systems was often studied in semi-dynamic models. Compared with static models, semi-dynamic models could obtain dynamic data about nutrient digestion and food structure changes orally and in the stomach and intestine [111]. Dynamic models are more complex, and fewer studies have used dynamic models to investigate in vitro digestion of DHA.

Factors influencing the in vitro digestion of DHA delivery systems include the type and concentration of digestive enzymes, pH value, digestion time, the secretion of digestive enzymes, bile salts, and manufacturing technology [45,112,113]. The in vitro digestibility of DHA microcapsules largely depended on the properties of wall materials. Wall materials made of phospholipid protein and maltodextrin could improve the release and digestibility of DHA in the stomach and small intestine. In the process of in vitro digestion of DHA microcapsule, amino acids and fatty acids were produced, which was due to the combined process of lipolysis and proteolysis [21]. The bioaccessibility of DHA in the stomach was also related to the stability of emulsion droplets during digestion. Pectin content and pH value in the stomach had an interactive effect on emulsion digestion. The addition of apple pectin could reduce the bioaccessibility of DHA. The presence of apple pectin and applesauce destabilized the emulsion, which also limited lipid digestibility and the bioaccessibility of DHA. In conclusion, a stable emulsion microstructure during gastric digestion enhanced the in vitro lipid digestibility and bioaccessibility of DHA [114].

### 3.2. In Vivo Digestion

The in vivo digestion model could better simulate the food digestion and absorption behavior in the body compared with the in vitro digestion model, from which the bioavailability of DHA could be determined. In vivo digestion is a dynamic process affected by peristalsis, gradual secretion of enzymes, and the residence time of food [115,116]. Mice or rat models are generally used for in vivo digestion experiments (Figure 3). Parthasarathi, Muthukumar, and Anandharamakrishnan [117] studied the biological properties of the prepared emulsions using male Wistar rats as the in vivo digestion model. The study of in vivo digestion could more directly explain the digestion, absorption, and bioavailability of food.

Microalgal oil contains a large amount of DHA and has a low hydrolysis rate when eaten. Therefore, the digestibility and absorption rate of microalgal oil are low, resulting in a low bioavailability of DHA [118]. However, a liposome delivery system, especially phospholipid, could improve DHA absorption. The hydrolysis of phospholipids may be the point for DHA release and absorption in the delivery system [119]. β-sitosterol was also used to prepare DHA liposomes, which could promote the release of DHA in the small intestine and improve the absorption of DHA [120]. The dietary substrate also affects the release of DHA during digestion. The true efficacy of food bioactive substances depends on four main steps, including gastrointestinal release, intestinal absorption, intestinal metabolism, and the effects on health. In this case, the dietary substrate could promote or prevent DHA release and dissolution during digestion to improve the bioavailability and availability of DHA. Pineda-Vadillo et al. [121] developed three DHA-rich egg products with the same ingredients but different structures, such as omelet, boiled egg, and mousse. Egg rolls were the most efficient system for increasing the bioavailability of DHA. The interaction between food substances and DHA delivery systems is also crucial in developing potentially effective DHA-rich foods.

## 4. Functionality of DHA in Delivery Systems

DHA is closely related to human health, but direct intake of DHA has a poor utilization rate. Delivery systems could improve the bioavailability and bioactivity of DHA, in which DHA could be absorbed more effectively and play the role of enhancing health (Figure 5). This section provides a detailed overview of the role of DHA delivery systems in impacting cardiovascular, diabetes, obesity, inflammation, visual, and neurological/brain diseases (Table 2), as well as the prospects for future research direction.

### 4.1. Effects on Cardiovascular Diseases

Cardiovascular diseases are the most important diseases endangering human health and life with high morbidity and mortality, but many are preventable. Obesity and chronic inflammation are thought to be the causes of many types of cardiovascular diseases [122]. Enrichment of omega-3 PUFAs, especially DHA, in the heart and brain are associated with a lower risk of cardiovascular disease. DHA has favorable effects on lipid regulation, vascular health, and heart rhythm, thus improving cardiovascular health. Delivery systems, especially microemulsions, are explored by many investigators to deliver biomolecules with greater efficiency. Increased bioavailability of DHA was observed when fish oils were delivered by microemulsion. Sugasini and Lokesh [23] prepared DHA-rich fish oil microemulsions with chitosan, acacia gum, whey protein, and lipoid, respectively. The fish oil microemulsions made with lipoid had the highest bioavailability, 68% higher than natural fish oil. The intestinal sac method and intubation were used to determine the DHA bioavailability of fish oil microemulsion and natural oil in rats. The results indicated that the DHA levels of microemulsions with the lipoid group were increased by 77% and 41% in the heart and brain lipids, respectively, compared to the natural fish oil group. DHA could be absorbed more efficiently in delivery systems. Lipid-binding materials such as lipoids could significantly enhance DHA levels in serum, liver, heart, and brain tissues. Thus, developing delivery systems to improve DHA bioavailability is a coveted direction to prevent cardiovascular diseases.

### 4.2. Effects on Visual and Neurological/Brain Development

DHA is the main lipid structure of the sensory and vascular retinas, an important part of infant vision development [123]. DHA has become a common ingredient in infant formula, but maintaining its stability and bioavailability remains challenging. Microencapsulated DHA tuna oil powder in infant formula could provide better DHA bioavailability. Fard et al. [25] found that blood DHA levels increased significantly in the microencapsulated group compared to non-encapsulated tuna oil. The high DHA level in the microencapsulated tuna oil group might be due to the microencapsulation improving the digestion, absorption, and effective protection of the tuna oil, which could offset the negative conditions (such as pH, oxidizing substances) and negative influence of other food ingredients in infants gastrointestinal system [124].

DHA is mainly located on the phospholipids in the neuron membrane, which is considered a neuroprotective agent against brain aging, neurodegenerative, and cerebrovascular diseases [125]. Evidence revealed that DHA might ameliorate cognitive decline and affect behavioral symptoms of serious neuropsychiatric disorders such as schizophrenia and depression [126]. Higher DHA intake is beneficial to increasing DHA in the brain, but it is relatively slow to increase DHA through dietary supplementation. LDL nanoparticles reconstituted with DHA had significant therapeutic value in the brain. Fluorescence imaging of rat brains indicated that DHA was transported into cells and then metabolized. LDL-DHA nanoparticles were easily incorporated into the phospholipid membrane of brain cells, which could enhance the DHA content in the brain [24]. The amyloid precursor protein (APP) and fluidity of the neuronal membrane play key roles in brain aging and neurodegeneration, and the fluidity of the membrane is crucial to regulating APP [127]. DHA liposome increased cell membrane fluidity and thereby promoted the production of sAPPα. Eckert et al. [128] prepared unilamellar DHA liposomes, which could protect DHA from oxidation and effectively transport DHA into cell membranes. DHA liposomes could maintain or restore physiological characteristics of cell membranes, which were necessary for neuroprotective sAPPα secretion and autocrine regulation of neuronal survival.

### 4.3. Effects on Diabetes Mellitus 

Diabetes mellitus is a human metabolic disease characterized by chronic inflammation of hyperglycemia and insulin resistance, one of the major afflictions to human health. Diabetes-related complications, including cardiovascular disease, retinopathy, kidney disease, and neuropathy, are important causes of increased morbidity and mortality in diabetes patients. Hyperglycemia has oxidative and pro-inflammatory properties, which could lead to complications of diabetes. High blood glucose could cause an increase in antioxidant capacity and induce the expression of pro-inflammatory cytokines such as tumor necrosis factor (TNF), interleukin-6 (IL-6), interleukin-8 (IL-8), monocyte chemotactic protein-1 (MCP-1), and nuclear factor-kappa B (NF-κB). Omega-3 PUFAs such as DHA have been proven to have beneficial effects in reducing oxidative stress and improving antioxidant systems. Fish oil emulsions increased the absorption of DHA compared to free fish oil, which could reduce oxidative stress induced by high blood glucose [26]. Hyperglycemia-induced inflammation is one of the key factors leading to diabetes complications, and monocytes/macrophages are the important part. Fish oil emulsions significantly increased the antioxidant capacity and SOD activity and decreased the release of TNF, IL-6, IL-8, and MCP-1 during the 72 h incubation of monocyte/macrophage U937 cells. The potential protective effect of fish oil emulsions against hyperglycemia-induced oxidative stress and inflammation may be due to the increased absorption of fatty acids. Fish oil emulsions reduce hyperglycemia-induced diabetes through the antioxidant and anti-inflammatory effects of DHA, but further studies are needed to examine whether DHA emulsions are associated with improved clinical symptoms in diabetes patients.

### 4.4. Anti-Obesity Effects

Obesity increases the risk of chronic non-communicable diseases such as cardiovascular disease, diabetes, and cancer, which can seriously damage people’s health and quality of life. Obesity is characterized by weight gain, excessive deposition of free fatty acids in adipose tissue, increased chronic inflammatory infiltration, and abnormal production of adipokines [129,130]. DHA has been proven to improve obesity-related metabolic disorders [2,131]. OLETF rats binge eat due to a lack of cholecystokinin receptors, which leads to obesity, fatty liver, hyperlipidemia, and diabetes. However, DHA could significantly reduce liver weight and hepatic triglyceride levels in OLETF rats to reduce obesity-induced fatty liver [27]. DHA-phosphatidylcholine prevented or alleviated obesity-related diseases by inhibiting adipogenesis, promoting lipolysis, and increasing adiponectin production in OLETF rats [67]. C57BL/6 mice fed with a high-fat diet (HFD) are often used as obesity models. DHA has the effect of lowering blood lipid, preventing the development of insulin resistance in obese animal models [9]. Compared with the HFD group of C57BL/6 mice, the body weight, liver lipid deposition, lipid peroxidation, and lipogenic-related proteins were significantly decreased in the DHA/EPA group [132]. DHA and the antioxidant hydroxytyrosol (HT) could improve the oxidative stress and mitochondrial dysfunction of obesity induced by HFD [133]. DHA-phospholipid (PL) had significant biological activity and the ability to reduce liver and serum cholesterol triglyceride levels [134]. DHA-PL inhibited weight gain in HFD-induced ICR mice and significantly decreased cholesterol and triglyceride levels. DHA could prevent liver steatosis and reduce the risk of atherosclerosis induced by obesity [135]. The anti-obesity mechanism of different DHA concentrations is different. 1% DHA inhibited adipogenesis by down-regulating G protein-coupled receptor 120 (GPR120). 4%DHA could up-regulate peroxisome proliferator-activated receptor-γ (PPARγ) to improve inflammatory infiltration and inhibit obesity [53]. DHA had an excellent anti-obesity effect, and its anti-obesity mechanism had been preliminary studied. However, there is no study on the effects of DHA delivery systems in anti-obesity. Whether DHA delivery systems could improve DHA absorption in obese patients and the mechanism of anti-obesity are needed to be studied.

### 4.5. Anti-Inflammatory Effects

Obesity was associated with chronic low-grade inflammation [136]. Inflammation was a response to the accumulation of lipids, which involved interactions between many cell types [137,138]. Chronic low-grade inflammation was associated with immune cells such as macrophages and T lymphocytes [139,140]. In obese patients, pro-inflammatory cytokine IL-6, TNF-α, MCP-1, and leptin increased, and the concentration of the anti-inflammatory cytokine adiponectin, IL-10, decreased [141,142]. DHA could reduce low-grade inflammation [143]. 3T3-L1 cells are commonly used to study adipose tissue obesity and inflammation. DHA significantly changed the adipokine secretion in 3T3-L1 cells and had an anti-inflammatory effect [28]. DHA decreased TNF-α, IL-1β in LPS-induced inflammation and mediated anti-inflammatory effects through the NF-κB signaling pathway [29,144]. DHA specifically enhanced anti-inflammatory IL-10 secretion and significantly inhibited the expression of IL-6, IL-1β, TNF-α in macrophages [29,30,145]. DHA also affects the treatment of inflammatory bowel disease through the NF-κB pathway to inhibit inflammation [146,147]. DHA has been demonstrated to improve inflammation induced by obesity and diabetes. However, there was no report on the anti-inflammation of DHA delivery systems. The anti-inflammatory effects of DHA delivery systems and their association with anti-obesity are the future research directions.

## 5. Food Applications of DHA Delivery Systems

DHA is a functional nutritional fortification used in infant milk powder and complementary foods. It is currently used in health care products, dairy products, bakery products, gel confectionery, and edible oil (Figure 6). The main source of DHA is seafood, but the related products often have unpleasant smells and tastes. Fish oil microcapsules are a good alternative to mask the fishy odor. Dairy products are good candidates for DHA fortification because of their high consumption frequency and ideal storage conditions. In the cheese-making process, fish oil microcapsules were added to the cheese matrix, which could improve the binding and retention of DHA in cheese, prolong the storage life, and provide a reference for the application of DHA in cheese [148,149]. Infant formula is also the main product of DHA application in dairy products. Compared with free DHA, microencapsulated DHA could provide improved bioavailability of DHA in infant formula. DHA microcapsule is an effective delivery system to improve the bioavailability of DHA in infant formula [25].

DHA could be efficiently delivered through baked goods, such as bread, nutritional bars, cereals, and cookies. Bread is a good product for DHA application, as bread is the staple food in most cultures worldwide [150]. The bread with the addition of 0.5% omega-3 fatty acids (rich in DHA) could be stabled in storage [151]. 25 or 50 mg of DHA per slice (32 g) was feasible for white bread [152]. Bread with microencapsulated DHA absorbed less water, which showed lower baking properties but had lower fishy flavor and peroxide [65,152,153]. However, microcapsule was the most feasible delivery system for DHA nutritional bar [154].

## 6. Conclusions and Future Trends

The importance of DHA in physiological function has been widely studied. The intake of DHA in appropriate amounts not only improves cardiovascular disease and diabetes but also benefits vision and neurological/brain, as well as inhibits obesity and associated inflammation. However, DHA has many disadvantages, including poor water solubility, high oxidation sensitivity, peculiar smell, and low bioavailability, which restrict its application in food. In recent years, progress has been made in using delivery systems to deliver DHA and improve its bioavailability, showing broad application prospects. Various delivery systems, such as microemulsions, nanoemulsions, multilayer emulsions, Pickering emulsions, liposomes, microcapsules, nanoparticles, and gels, are available to improve the encapsulation, protection, release, and bioavailability of DHA. Biological characteristics, chemical stability, compatibility of food substrate, and economic feasibility should be considered when selecting DHA delivery systems. Microcapsules reduce the contact between DHA and active substances in food and gastrointestinal fluids (such as acids and enzymes), allowing DHA to be better released and absorbed in the intestine. Overcoming the physical and chemical degradation of delivery systems in the gastrointestinal tract could be achieved by using microcapsules.

However, it should be pointed out that the delivery systems discussed above could not only provide guidance for DHA in food applications but also stimulate the delivery of other substances. Although some studies have revealed the efficacy of DHA in the in vitro models, little attention has been paid to the DHA delivery systems in vivo, which makes it difficult to demonstrate their effectiveness accurately. In addition, the safety and sensory quality evaluation of DHA delivery systems should be further verified. Furthermore, natural functional ingredients (such as protein and polysaccharides) are considered eco-friendly and safe, which are worthwhile for further development of DHA delivery systems based on natural ingredients with multiple functions and superior performance. To design and select a suitable DHA delivery system, attention should be focused on overcoming the instability of DHA in processing, storage, and transportation, improving the bioavailability of DHA, and developing commercial values in the food field.

## Figures and Tables

**Figure 1 foods-11-02685-f001:**
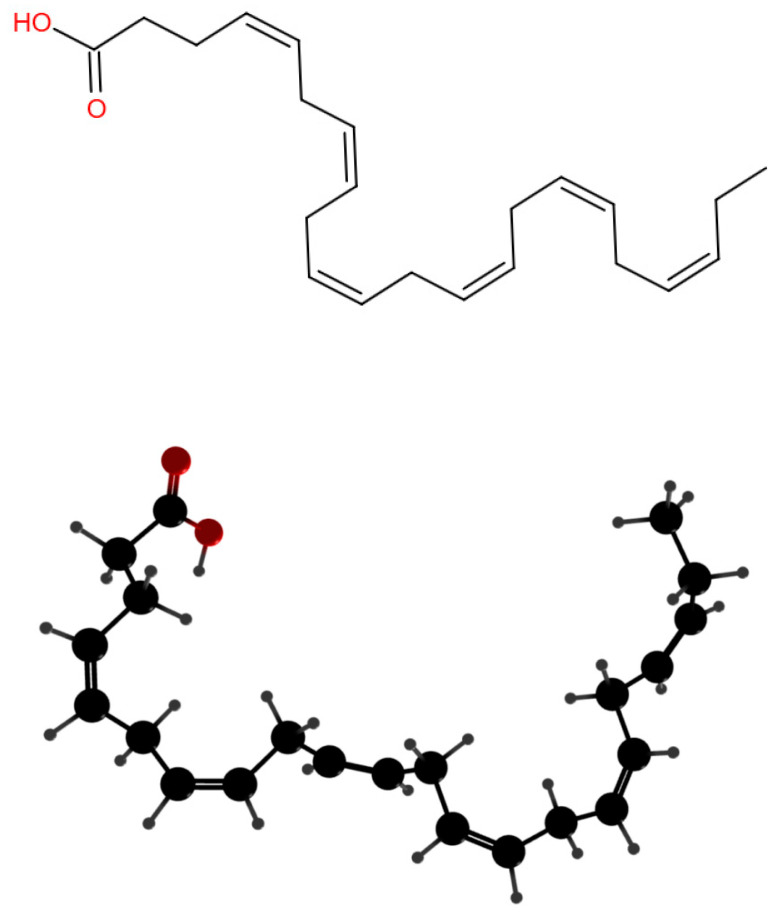
DHA molecular structure.

**Figure 3 foods-11-02685-f003:**
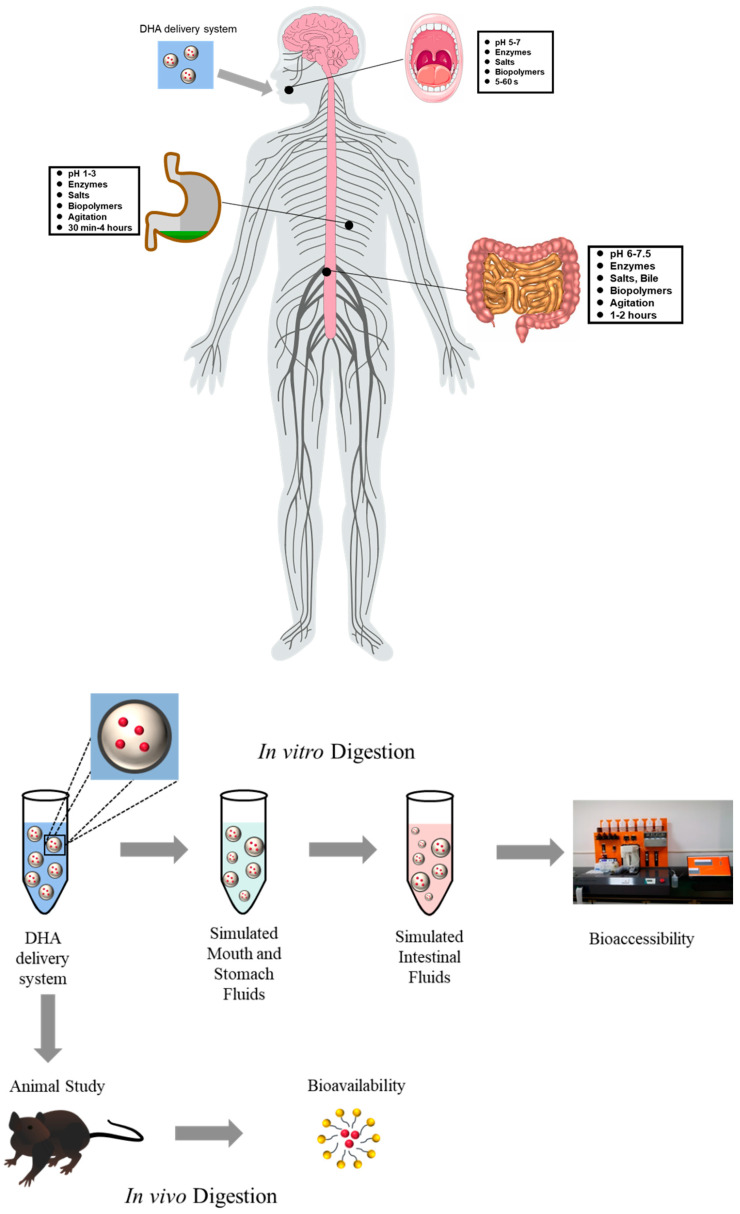
In vitro and in vivo digestion model of DHA delivery systems.

**Figure 4 foods-11-02685-f004:**
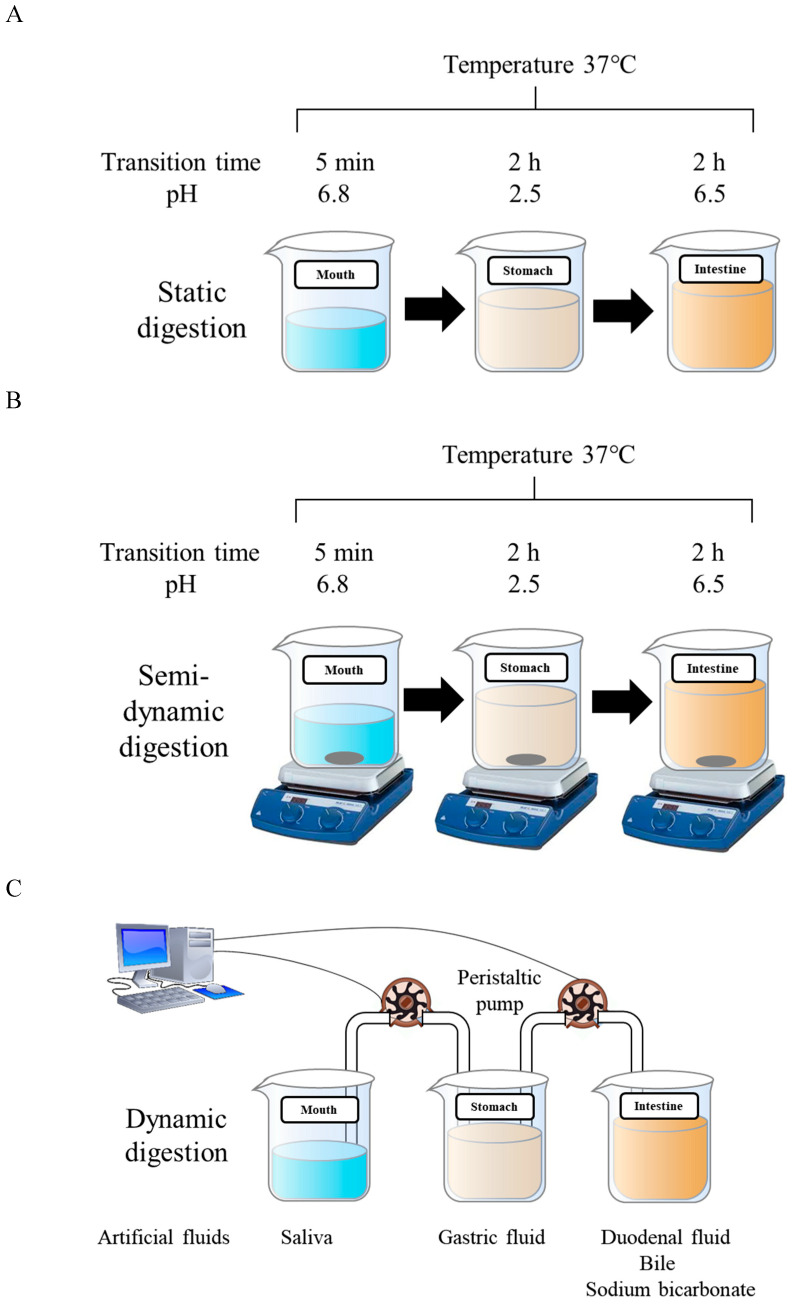
Static digestion (**A**), semi-dynamic digestion (**B**), and dynamic digestion (**C**) models of DHA delivery systems.

**Figure 5 foods-11-02685-f005:**
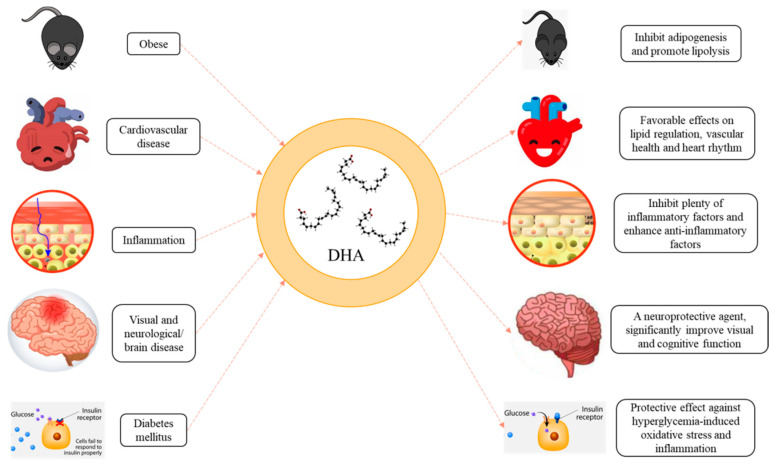
Functionality of DHA in delivery systems.

**Figure 6 foods-11-02685-f006:**
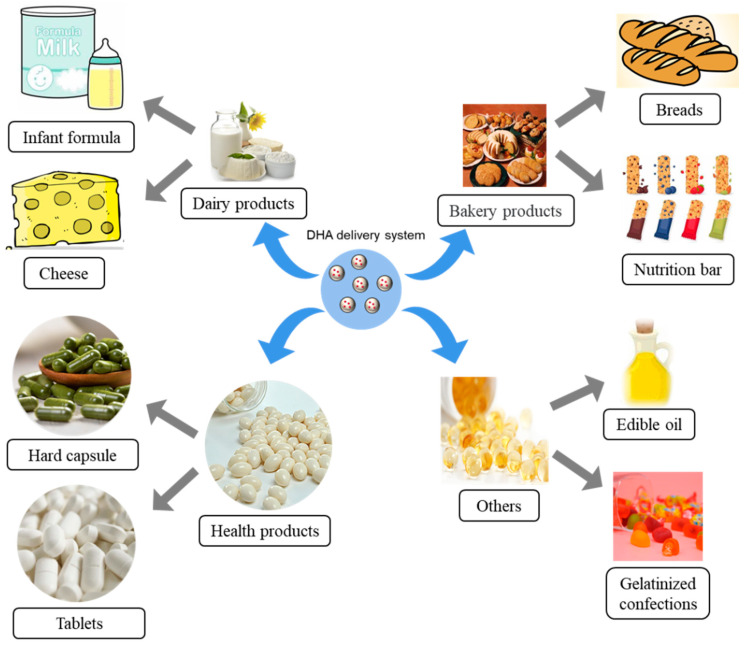
Food applications of DHA delivery systems.

## Data Availability

The data presented in this study are available on request from the corresponding author.

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
