# Peer review of "Docosahexaenoic Acid Delivery Systems, Bioavailability, Functionality, and Applications: A Review"

_foods, 2022, doi:10.3390/foods11172685_

Round 1

Reviewer 1 Report

Lv and Xu have reviewed DHA delivery systems, bioavailability, functionality and applications. The topic of this review is interesting and the need for this review is timely. However, there are several issues as shown below need to be addressed before this paper could be considered for publication in Foods:

1.      Consider revising the phrase “maximize the remain of DHA” in the abstract.

2.      In section 2.2, specify the difference between nanoemulsion and microemulsion.

3.      At the end of section 2.2, a paragraph comparing different DHA delivery system recently reported, highlight the advantages and disadvantages of them and which ones seem best.

4.      A table can be developed for the section 4 based on the published articles reviewed in this section.

5.      In food application section 5, the authors have not discussed articles published in the recent years. Some more recently published article should be searched for and discussed.

6.      Some basic physicochemical data should be added in the form of a table for DHA.

7.      Figure 4, static digestion, semi-dynamic digestion and dynamic digestion should be labelled as A, B, and C in the caption and figures.

8.      Figures 3 and 5 should be enlarged to facilitate better visual clarity and readability.

Reviewer 2 Report

In the Introduction Section, The Authors pointed very clear the main properties of DHA, its benefits for human health, and also the main problems associated with its administration.

In the Section 2, the Authors clearly detailed the main delivery systems of DHA, highlighting the advantages for each of them as carrier for DHA, focusing especially on the improving the bioavailability, the stability, the solubility and masking of the smell.

In the next section, the Authors paid attention to the digestion of DHA in delivery systems, studied with the in vitro and in vivo methods.

An important part of this review is dedicated to functionality of DHA in delivery systems. The Authors exhaustively presented the role played by DHA delivery systems in different diseases.

Beside the application of DHA delivery systems in different diseases, the Authors also provided useful information related to the food applications of DHA delivery systems.

The purpose of this review, focused on different delivery systems of DHA and their functionality in different diseases, as well as on the applications of DHA delivery systems in food, is well supported by a documented information. Also, this article is sustained by an important number of references, correctly cited and up to date.

 Concerning the above manuscript, I have some remarks related to:

1. In my opinion, in Title please replace the abbreviation DHA with the full name.

2. Please insert Table 1 with body article.

3. English minor revision and spelling corrections are required.

Round 2

Reviewer 1 Report

The authors have satisfactorily addressed all the comments raised by reviewers and therefore I recommend acceptance of this article for publication in Foods.